# Artificial intelligence and leukocyte epigenomics: Evaluation and prediction of late-onset Alzheimer's disease

Ray O. Bahado-Singh[1], Sangeetha Vishweswaraiah[1], Buket Aydas[2], Ali Yilmaz[1], Raghu P. Metpally[3], David J. Carey[3], Richard C. Crist[4], Wade H. Berrettini[4], George D. Wilson[5], Khalid Imam[6], Michael Maddens[6], Halil Bisgin[7], Stewart F. Graham[1], Uppala Radhakrishna[1]*

1 Department of Obstetrics and Gynecology, Oakland University-William Beaumont School of Medicine, Royal Oak, Michigan, United States of America, 2 Department of Healthcare Analytics, Meridian Health Plans, Detroit, Michigan, United States of America, 3 Department of Molecular and Functional Genomics, Geisinger, Danville, Pennsylvania, United States of America, 4 Department of Psychiatry, Perelman School of Medicine, University of Pennsylvania, Pennsylvania, Pennsylvania, United States of America, 5 Department of Radiation Oncology, Oakland University-William Beaumont School of Medicine, Rochester, Michigan, United States of America, 6 Department of Internal Medicine, Oakland University-William Beaumont School of Medicine, Rochester, Michigan, United States of America, 7 Department of Computer Science, University of Michigan, Flint, Michigan, United States of America

* radhakrishna.uppala@unige.ch

## Abstract

We evaluated the utility of leucocyte epigenomic-biomarkers for Alzheimer's Disease (AD) detection and elucidates its molecular pathogeneses. Genome-wide DNA methylation analysis was performed using the Infinium MethylationEPIC BeadChip array in 24 late-onset AD (LOAD) and 24 cognitively healthy subjects. Data were analyzed using six Artificial Intelligence (AI) methodologies including Deep Learning (DL) followed by Ingenuity Pathway Analysis (IPA) was used for AD prediction. We identified 152 significantly (FDR $p < 0.05$) differentially methylated intragenic CpGs in 171 distinct genes in AD patients compared to controls. All AI platforms accurately predicted AD with AUCs $\geq 0.93$ using 283,143 intragenic and 244,246 intergenic/extragenic CpGs. DL had an AUC = 0.99 using intragenic CpGs, with both sensitivity and specificity being 97%. High AD prediction was also achieved using intergenic/extragenic CpG sites (DL significance value being AUC = 0.99 with 97% sensitivity and specificity). Epigenetically altered genes included *CR1L* & *CTSV* (abnormal morphology of cerebral cortex), *S1PR1* (CNS inflammation), and *LTB4R* (inflammatory response). These genes have been previously linked with AD and dementia. The differentially methylated genes *CTSV* & *PRMT5* (ventricular hypertrophy and dilation) are linked to cardiovascular disease and of interest given the known association between impaired cerebral blood flow, cardiovascular disease, and AD. We report a novel, minimally invasive approach using peripheral blood leucocyte epigenomics, and AI analysis to detect AD and elucidate its pathogenesis.

**Data Availability Statement:** All relevant data are within the manuscript and its Supporting Information files.

**Funding:** The funder provided support in the form of salaries for authors [BA], but did not have any additional role in the study design, data collection, and analysis, decision to publish, or preparation of the manuscript. The specific roles of these authors are articulated in the 'author contributions' section.

**Competing interests:** The authors have read the journal's policy and have the following competing interest: BA is a paid employee of Meridian HealthComms Ltd. There are no patents, products in development or marketed products associated with this research to declare. This does not alter our adherence to PLOS ONE policies on sharing data and materials.

## Introduction

Alzheimer's Disease (AD) is the most common form of age-related dementia, accounting for 60–80% of such cases [1]. The disorder causes a wide range of significant mental and physical disabilities, with profound behavioral changes and progressive impairment of social skills. Globally in 2015, nearly 47 million individuals suffered from AD and it is projected that 75 million will be affected by 2030, with a further rise to 131 million by 2050 [2]. The World Health Organization has therefore declared AD a global health priority [3].

AD is a complex disorder influenced by environmental and genetic factors [4,5]. Many studies have investigated the genetic basis for both early-onset AD (EOAD) and late-onset AD (LOAD) [6,7]. Genome-wide association studies (GWAS) [8] have identified several LOAD-associated risk loci [9] proliferation in peripheral blood leukocytes including in T-lymphocytes [10], B-lymphocytes [11], polymorphonuclear leucocytes [12], monocytes, and macrophages [13] have been reported. DNA methylation plays an important role in Alzheimer's disease [14–16]. Leukocyte DNA methylation from CpG-based biomarker analyses was used for early detection of many diseases, including our recently published brain disorders cerebral palsy [17], autism [18], and concussion [19]. However, the genome-wide blood DNA methylation-based molecular mechanisms that contribute to the pathogenesis of AD remain still largely unknown.

Artificial Intelligence (AI) is rapidly transforming modern life in areas as diverse as face recognition and robotics. Machine Learning (ML) is a branch of AI that focuses on computer learning and adapting from a set of data with which it has been presented. ML involves learning by computers that require no or only minimal explicit programming by humans. An area of interest given the geometric expansion of medical data is the use of ML for the detection and diagnosis of various diseases [20]. ML has been reported to be superior to conventional statistical approaches for prediction such as logistic regression and Cox proportional hazard model-based analysis [21] when interrogating mega-data. Challenges with classical statistical techniques include but are not limited by the requirement for an assumption of independence between predictors and risk of overfitting and collinearity when a large number of variables are analyzed. Deep Learning (DL) is the latest developing branch of ML. DL uses multi-layered neural networks that are modeled after neural networks in the brain of animals, to learn essential tasks. Thus, with minimal or no explicit human programming (unsupervised), the computer can learn intricate patterns from complex data matrices. When subsequently exposed to a new data set, it can classify and make precise predictions based on past experiences. With DL, between the input (raw data) and output (i.e. completed task e.g. group classification) layer of 'neurons,' there are multiple hidden layers that enhance the ability to handle tasks of increasing complexity. DL more closely mimics the intellectual function of the cerebral cortex. There is an increasing interest in using DL in the analysis of biologic big-data such as genomics [22,23] to understand and accurately predict diseases. We have recently published using AI/ML-based technologies of epigenomic [17] and metabolomics [24–26] data for accurate disease prediction. In the present study, we used DL and other commonly used ML platforms combined with genome-wide DNA methylation analysis of leucocytes DNA for AD detection/prediction. The term 'prediction' is used here in a cross-sectional as opposed to a temporally longitudinal sense since the samples were not obtained before the development of AD. To further explore the molecular mechanisms of LOAD, we used the Ingenuity Pathway Analysis (IPA).

## Materials and methods

Institutional Review Board (IRB) approval was provided by William Beaumont Hospital, Royal Oak MI, USA (IRB#2014–038). Written consent was obtained from all participants and

their legally authorized representatives when applicable. The diagnosis of AD in these live subjects was made using the published criteria of NINCDS-ADRDAj [27]. Demographic and clinical data were extracted from the medical records (S1 Table) and compared between AD and control groups. Genomic DNA was extracted from whole blood samples using the Gentra Puregene Blood Kit (Qiagen) according to the manufacturer's protocol. Approximately 500 ng of genomic DNA was extracted from each of the 48 samples, which subsequently were bisulfite converted using the EZ DNA Methylation-Direct Kit (Zymo Research, Orange, CA) per the manufacturer´s protocol and processed according to Illumina protocols. Bisulfite conversion was performed in a PCR cycling protocol (16 x 95˚C for 30 sec, 50˚C for 60 min) and then held at 4˚C.

## Genome-wide methylation scan using the Infinium MethylationEPIC array BeadChips

The Infinium MethylationEPIC array (Illumina, Inc., California, USA) contains probes for >850,000 CpGs per sample. All 48 samples were processed together to minimize batch effects. This is further elucidated in the Supplementary Methods. This section also includes validation results using pyrosequencing along with primer sequences.

## Statistical and bioinformatic analysis

Differential methylation was determined by comparing the ß-values per individual nucleotide at each cytosine 'CpG' locus between AD subjects and controls. The p-value for the methylation difference between AD and control groups at each locus was calculated as previously described [28]. Probes associated with X and Y chromosomes were removed to negate any bias caused by gender differences. Further detailed statistical and bioinformatic analyses are described in the Supplementary section.

## Artificial Intelligence (AI) analysis

AI analysis was performed as previously described by our group [29], using a combination of CpG sites from different genes. A total of six different AI platforms including Deep Learning (DL) were evaluated. Each CpG locus used as a marker displayed significant differential methylation in AD defined as FDR p-value <0.05. The methylation β-values were logged and autoscaled using their standard deviation before quantile normalization to minimize sample to sample difference. Standard techniques were used with DL including adjustments by the program of weights (strength of the connection between 'neurons') and biases (an additional parameter or constant) and backpropagation—all of which helps to optimize the accuracy of the output or results. Softmax classifier was used to assign new labels to the samples. To tune the parameters of the DL model, the h2o package in the R module was used [30,31]. For the sake of comparison, standard logistic regression algorithms for AD prediction were also performed and detailed later in the manuscript.

## Other machine learning algorithms

We compared the performance of DL to five other commonly used machine learning algorithms: Support Vector Machine (SVM), Generalized Linear Model (GLM), Prediction Analysis for Microarrays (PAM), Random Forest (RF), and Linear Discriminant Analysis (LDA) [30,32]. A comprehensive explanation of the AI methodology is provided in the Supplementary Section.

## Bootstrapping

We also performed bootstrapping as alternative 10-fold cross-validation and compared the new results with that based on 10-fold CV. The bootstrap method involves iteratively resampling a dataset with replacement. Instead of only estimating our statistic once on the complete data, this can be performed many times on a re-sampling (with replacement) of the original sample. We repeated this re-sampling 100 times and averaged the results.

## Results

A total of 24 LOAD subjects and 24 cognitively healthy controls were used in this study. Selected clinical and demographic characteristics were compared between AD and control groups (S1 Table). There were no significant differences in age, gender, and common cardio-vascular diseases between groups. There was a higher percentage of females in both the study and control groups consistent with LOAD demographics; however, gender was not significantly (p = 0.53) different between groups. The MMSE (mini-mental status exam) is a psychological test commonly administered to screen for AD. As expected, the MMSE test score was significantly lower in the AD than in the control group (p-$1.54 \times 10^{-7}$). A comparison of the methylation profiles between AD and control subjects revealed 152 differentially methylated intragenic CpG sites (FDR p<0.05 and fold change $\geq$1.5) associated with 171 unique genes. We validated two randomly chosen CpGs by pyrosequencing and confirmed the top-ranking hits in the whole blood DNA of our cohort samples. These analyses revealed similar methylation data like those from the Illumina Infinium MethylationEPIC arrays, indicating that the initial methylation changes were not artifacts. 33 intragenic CpG sites met the GWAS stringent p-value thresholds i.e. p<$5 \times 10^{-8}$ (Table 1). A total of 17 separate intragenic CpG sites had moderate to good individual predictive accuracy (AUC $\geq$ 0.75) for AD detection based on methylation levels. An additional 119 CpG markers displaying significant methylation differences (FDR p-value <0.05) between AD and controls are presented in S2 Table. Both hyper- (66.4%) and hypomethylation (33.6%) were observed among intragenic CpG sites in the AD cases.

A prior report found significant differential methylation of intergenic/extragenic sites in the leukocyte genome in AD [33] which correlated with the performance on the MMSE. Based on this we also evaluated the methylation changes in intergenic/extragenic CpG sites for AD prediction. Highly significant differences in CpG methylation were observed for multiple intergenic/extragenic sites throughout the genome. This was observed when using different thresholds to define statistical significance: A total of 1524 intergenic/extragenic CpGs with FDR p-value <0.05 and 103 intergenic/extragenic CpGs using a stringent threshold (p<$5 \times 10^{-8}$) were identified [34]. The top 25 intergenic/extragenic markers for AD prediction using the different statistical thresholds mentioned above are listed in Tables 2 and 3.

Principal Component Analysis (PCA) and Partial Least Square Discriminant Analyses (PLS-DA) confirmed significant segregation of AD cases from controls using intragenic CpG methylation markers (Fig 1). Permutation testing indicated that the separation observed between the AD and control groups was highly statistically significant (p<$5 \times 10^{-8}$) and not likely due to chance.

For most of our analyses, conventional statistical tools were used to first identify high performing individual markers as indicated by AUC or FDR p-value thresholds, and these subsets of markers were then subjected to AI analyses. This approach has the advantage of reducing AI computing time and therefore costs. Prior publications suggest however that ML approaches might be superior to conventional statistical methods such as logistic regression

**Table 1. Top 33 differentially methylated CpG markers—(Gene IDs, chromosome location, fold change, AUC, and percentage of methylation difference for each CpG).**

| Target ID | CHR | Gene | FDR p-Val | Fold change | AUC | CI | | % Methylation | | % Methylation difference |
|-----------|-----|------|-----------|-------------|-----|-------|-------|-------|---------|--------------------------|
| | | | | | | Lower | Upper | Cases | Control | |
| cg20008763 | 19 | ZNF667 | 2.75071E-41 | 1.57 | 0.66 | 0.51 | 0.81 | 44.46 | 28.33 | 16.13 |
| cg25755428 | 19 | MRI1 | 4.25655E-41 | 1.57 | 0.65 | 0.50 | 0.81 | 43.40 | 27.67 | 15.73 |
| cg21353034 | 12 | VPS33A | 1.11479E-38 | 1.93 | 0.65 | 0.50 | 0.81 | 21.49 | 11.13 | 10.36 |
| cg05706624 | 17 | WSCD1 | 3.0837E-38 | 1.81 | 0.68 | 0.53 | 0.83 | 20.84 | 11.53 | 9.31 |
| cg12949483 | 15 | TMEM85 | 4.00337E-38 | 1.66 | 0.68 | 0.53 | 0.83 | 22.62 | 13.58 | 9.03 |
| cg26856451 | 2 | THAP4 | 1.35084E-37 | 2.25 | 0.74 | 0.60 | 0.88 | 13.92 | 6.19 | 7.73 |
| cg26340737 | 6 | RNF5P1; RNF5; AGPAT1 | 1.52469E-37 | 2.84 | 0.60 | 0.44 | 0.76 | 11.73 | 4.13 | 7.60 |
| cg04515524 | 19 | PLVAP | 1.15699E-30 | 0.39 | 0.75 | 0.61 | 0.89 | 8.28 | 21.08 | -12.80 |
| cg02356786 | 1 | LOC731275 | 3.59905E-21 | 0.48 | 0.71 | 0.57 | 0.86 | 9.95 | 20.76 | -10.81 |
| cg05841700 | 1 | PM20D1 | 4.61313E-19 | 0.65 | 0.62 | 0.46 | 0.78 | 26.67 | 40.75 | -14.08 |
| cg16259859 | 1 | ZBTB8A | 2.79863E-17 | 0.60 | 0.66 | 0.51 | 0.82 | 18.09 | 29.95 | -11.86 |
| cg08829299 | 11 | ATHL1 | 2.25432E-16 | 0.62 | 0.67 | 0.52 | 0.83 | 18.59 | 30.21 | -11.63 |
| cg10326472 | 6 | MYB | 8.45895E-14 | 1.50 | 0.69 | 0.54 | 0.84 | 29.08 | 19.34 | 9.74 |
| cg00613827 | 1 | CR1L | 2.86297E-12 | 0.52 | 0.61 | 0.45 | 0.77 | 7.96 | 15.32 | -7.36 |
| cg07509935 | 14 | LTB4R; CIDEB | 4.66298E-12 | 0.53 | 0.68 | 0.52 | 0.83 | 8.45 | 15.88 | -7.43 |
| cg08611411 | 1 | LOR | 2.87911E-11 | 1.97 | 0.55 | 0.38 | 0.71 | 12.41 | 6.30 | 6.12 |
| cg18157505 | 1 | PTPRC | 5.51496E-11 | 1.71 | 0.63 | 0.47 | 0.78 | 16.62 | 9.73 | 6.89 |
| cg27119318 | 21 | WRB | 1.0259E-10 | 0.61 | 0.69 | 0.54 | 0.84 | 12.64 | 20.66 | -8.02 |
| cg01819759 | 13 | RNF219 | 1.02625E-10 | 1.54 | 0.61 | 0.45 | 0.77 | 22.24 | 14.45 | 7.80 |
| cg01887804 | 15 | IVD | 1.62867E-10 | 1.70 | 0.65 | 0.50 | 0.81 | 16.24 | 9.54 | 6.70 |
| cg23623880 | 1 | MACF1 | 2.95897E-10 | 1.52 | 0.69 | 0.54 | 0.84 | 22.53 | 14.83 | 7.70 |
| cg07469467 | 12 | APAF1 | 4.81236E-10 | 0.58 | 0.63 | 0.47 | 0.78 | 9.82 | 16.88 | -7.06 |
| ch.15.658653F | 15 | TMOD2 | 7.3646E-10 | 0.55 | 0.64 | 0.48 | 0.80 | 7.81 | 14.28 | -6.47 |
| cg17160660 | 8 | MYC | 1.15589E-09 | 1.89 | 0.72 | 0.58 | 0.87 | 12.06 | 6.39 | 5.67 |
| cg16251399 | 6 | GUSBL2 | 1.28686E-09 | 0.47 | 0.65 | 0.50 | 0.81 | 5.02 | 10.62 | -5.60 |
| cg17578275 | 2 | ADAM17 | 1.29935E-09 | 0.60 | 0.67 | 0.52 | 0.82 | 10.93 | 18.09 | -7.16 |
| cg05800065 | 4 | NSG1 | 1.93339E-09 | 1.99 | 0.71 | 0.57 | 0.86 | 10.78 | 5.43 | 5.36 |
| cg19819404 | 4 | ZNF718 | 1.99376E-09 | 1.61 | 0.68 | 0.53 | 0.83 | 17.41 | 10.80 | 6.61 |
| cg00106073 | 1 | LMNA | 7.50554E-09 | 1.93 | 0.69 | 0.53 | 0.84 | 10.85 | 5.62 | 5.23 |
| cg24368383 | 1 | MIB2 | 1.5593E-08 | 2.44 | 0.62 | 0.46 | 0.78 | 7.62 | 3.12 | 4.50 |
| cg02722613 | 4 | SEPSECS | 1.69962E-08 | 0.63 | 0.60 | 0.43 | 0.76 | 11.38 | 18.19 | -6.80 |
| cg00853940 | 2 | TRPM8 | 2.05842E-08 | 1.50 | 0.68 | 0.52 | 0.83 | 20.05 | 13.33 | 6.73 |
| cg14304349 | 11 | TRIM6 | 3.15543E-08 | 0.49 | 0.59 | 0.43 | 0.76 | 4.87 | 9.96 | -5.10 |

analysis for group discrimination and risk prediction. [35]. Thus, direct AI analysis of the entire CpG data-space may improve AD prediction.

Using the direct AI analysis approach improved the predictive accuracy. Direct analysis of 283,143 individual intragenic markers CpGs improved predictive accuracy (Table 4) as did a direct analysis of 244,246 intergenic (extragenic) CpGs, (Table 5). Almost all ML platforms yielded a high predictive accuracy with an AUC $\geq$0.93. In the case of Deep Learning, using direct analysis of the intragenic markers, we observed AUC's = 0.992 with both sensitivities and specificities of $\geqq$97% for AD prediction, respectively (Table 4). For the intergenic (extragenic) markers, direct AI analysis (Table 5) yielded an AUC = 0.999 for DL with both sensitivities and specificities of = 97.5% for AD prediction. Our findings suggest that direct AI analysis of the raw methylation data could perform as well as or even further improve predictive

**Table 2. Top 25 intergenic/extragenic markers*.**

| Target ID | FDR p-Val | Fold change | AUC | CI | | % Methylation | | % Methylation difference |
|---|---|---|---|---|---|---|---|---|
| | | | | Lower | Upper | Cases | Control | |
| cg04299067 | 1.11E-14 | 1.31 | 0.79 | 0.66 | 0.92 | 49.58 | 37.83 | 11.76 |
| cg02147364 | 1.94E-14 | 0.4 | 0.77 | 0.64 | 0.91 | 4.49 | 11.35 | -6.87 |
| cg15711973 | 3.93E-10 | 0.86 | 0.77 | 0.63 | 0.9 | 59.61 | 69.68 | -10.07 |
| cg23332294 | 4.00E-07 | 1.13 | 0.76 | 0.63 | 0.9 | 69.15 | 61.27 | 7.88 |
| cg11166167 | 4.70E-06 | 0.91 | 0.78 | 0.65 | 0.91 | 68.17 | 75.27 | -7.1 |
| cg22680058 | 5.03E-05 | 1.61 | 0.77 | 0.64 | 0.91 | 10.9 | 6.77 | 4.14 |
| cg05293897 | 7.50E-05 | 0.83 | 0.77 | 0.64 | 0.91 | 35.64 | 43.18 | -7.54 |
| cg00614617 | 0.000121 | 1.15 | 0.8 | 0.68 | 0.93 | 53.81 | 46.71 | 7.1 |
| cg12269972 | 0.000145 | 0.81 | 0.76 | 0.62 | 0.9 | 30.32 | 37.36 | -7.04 |
| cg08343820 | 0.000357 | 0.92 | 0.78 | 0.65 | 0.91 | 69.33 | 75.21 | -5.87 |
| cg06336897 | 0.000397 | 0.9 | 0.79 | 0.66 | 0.92 | 60.13 | 66.73 | -6.6 |
| cg23980569 | 0.000405 | 0.89 | 0.81 | 0.68 | 0.93 | 56.04 | 62.86 | -6.82 |
| cg16219773 | 0.000518 | 0.93 | 0.76 | 0.63 | 0.9 | 71.94 | 77.43 | -5.5 |
| cg13699771 | 0.000844 | 1.12 | 0.76 | 0.62 | 0.9 | 59.28 | 52.91 | 6.37 |
| cg24328568 | 0.001267 | 0.9 | 0.8 | 0.67 | 0.92 | 56.64 | 63.03 | -6.39 |
| cg11122899 | 0.003358 | 0.83 | 0.79 | 0.66 | 0.92 | 26.86 | 32.53 | -5.67 |
| cg22509132 | 0.019164 | 0.93 | 0.76 | 0.63 | 0.9 | 65.3 | 70.13 | -4.83 |
| cg00280895 | 0.023204 | 1.11 | 0.77 | 0.64 | 0.91 | 51.94 | 46.75 | 5.19 |
| cg26041076 | 0.024238 | 0.73 | 0.81 | 0.69 | 0.94 | 8.46 | 11.53 | -3.08 |
| cg06858692 | 0.026431 | 0.92 | 0.76 | 0.63 | 0.9 | 57.32 | 62.41 | -5.09 |
| cg08895936 | 0.028385 | 1.03 | 0.77 | 0.63 | 0.9 | 87.82 | 84.95 | 2.87 |
| cg12688483 | 0.031136 | 1.59 | 0.76 | 0.63 | 0.9 | 6.38 | 4.02 | 2.36 |
| cg25906247 | 0.033333 | 1.43 | 0.77 | 0.63 | 0.9 | 8.68 | 6.06 | 2.61 |
| cg00521380 | 0.036657 | 0.96 | 0.8 | 0.67 | 0.92 | 77.56 | 81.17 | -3.61 |
| cg23694799 | 0.043606 | 0.94 | 0.84 | 0.72 | 0.95 | 64.7 | 69.18 | -4.49 |

*Methylation difference defined as FDR p-value <0.05*.

performance compared to analysis based on high performing individual CpG loci determined by conventional statistical approaches (see below).

As noted above we looked at the predictive performance of AI-based analysis of DNA methylation levels in intragenic and intergenic/extragenic CpG sites using individual markers that achieved different significance thresholds for AD prediction. High predictive accuracies were also achieved with these CpG markers using significance threshold FDR p-value<0.05 (S3 and S4 Tables) followed by the stringent significance threshold p-value <5X10$^{-8}$ (S5 and S6 Tables). DL appears to perform slightly better than other ML platforms however much larger case numbers would be required to assess this definitively. Increasing the number of predictors to 10 or 20 CpG loci did not appear to meaningfully improve predictive performance over the use of only 5 predictors. Similarly bootstrapping (1,000 samplings) yielded essentially similar results.

## Logistic regression analysis

We further investigated the performance of conventional logistic regression for comparison purposes. The methylation status of a combination of CpG markers: cg04515524, cg00613827, cg02356786, and cg07509935 was a good predictor of AD. The following performance was

**Table 3. Top 25 intergenic/extragenic markers: Genome-wide significance threshold[*].**

| Target ID | p-Val | Fold change | AUC | CI | | % Methylation | | % Methylation difference |
|---|---|---|---|---|---|---|---|---|
| | | | | Lower | Upper | Cases | Control | |
| rs4331560 | 3.39589E-45 | 1.84 | 0.68 | 0.53 | 0.83 | 51.47 | 27.97 | 23.50 |
| rs5926356 | 2.7196E-42 | 1.53 | 0.60 | 0.44 | 0.76 | 52.13 | 33.98 | 18.15 |
| rs10936224 | 1.42552E-40 | 1.35 | 0.59 | 0.43 | 0.76 | 56.30 | 41.68 | 14.62 |
| rs1040870 | 1.82909E-40 | 1.50 | 0.58 | 0.42 | 0.74 | 43.36 | 28.96 | 14.39 |
| cg11468315 | 4.90383E-40 | 1.67 | 0.66 | 0.50 | 0.81 | 33.67 | 20.22 | 13.46 |
| cg27128435 | 8.16168E-40 | 1.32 | 0.74 | 0.60 | 0.88 | 53.64 | 40.67 | 12.97 |
| rs348937 | 1.81674E-39 | 1.35 | 0.61 | 0.45 | 0.77 | 47.37 | 35.19 | 12.19 |
| cg00727777 | 9.80978E-39 | 1.15 | 0.58 | 0.42 | 0.74 | 79.50 | 69.01 | 10.50 |
| cg27055313 | 2.16905E-38 | 1.75 | 0.75 | 0.62 | 0.89 | 22.50 | 12.82 | 9.68 |
| cg19775763 | 1.4397E-37 | 1.09 | 0.62 | 0.46 | 0.78 | 90.66 | 83.00 | 7.66 |
| cg19432688 | 8.03822E-37 | 1.07 | 0.53 | 0.36 | 0.69 | 93.51 | 87.76 | 5.75 |
| rs264581 | 1.75175E-28 | 0.52 | 0.63 | 0.47 | 0.79 | 16.87 | 32.17 | -15.30 |
| rs1495031 | 9.34546E-28 | 0.63 | 0.66 | 0.51 | 0.82 | 42.09 | 67.03 | -24.93 |
| rs2032088 | 1.11017E-26 | 0.60 | 0.60 | 0.44 | 0.76 | 24.32 | 40.75 | -16.42 |
| rs6982811 | 1.87656E-26 | 0.60 | 0.61 | 0.45 | 0.77 | 24.88 | 41.32 | -16.44 |
| rs6626309 | 4.51648E-26 | 0.72 | 0.60 | 0.44 | 0.76 | 44.25 | 61.55 | -17.31 |
| cg27438152 | 9.37531E-21 | 0.84 | 0.67 | 0.51 | 0.82 | 65.54 | 78.42 | -12.88 |
| cg23155965 | 7.14189E-19 | 0.91 | 0.68 | 0.53 | 0.83 | 81.52 | 90.01 | -8.49 |
| cg16097834 | 2.76613E-18 | 0.87 | 0.50 | 0.34 | 0.67 | 73.12 | 83.68 | -10.56 |
| rs2208123 | 8.75703E-17 | 0.80 | 0.64 | 0.49 | 0.80 | 52.83 | 66.31 | -13.49 |
| cg25556225 | 2.17724E-16 | 0.86 | 0.73 | 0.59 | 0.88 | 66.98 | 78.32 | -11.33 |
| cg00224807 | 3.28345E-16 | 0.90 | 0.62 | 0.46 | 0.78 | 77.81 | 86.73 | -8.92 |
| cg03192273 | 1.66265E-15 | 0.61 | 0.63 | 0.47 | 0.78 | 17.06 | 28.01 | -10.95 |
| rs7746156 | 4.86899E-15 | 1.29 | 0.57 | 0.41 | 0.73 | 53.95 | 41.92 | 12.03 |
| rs5987737 | 4.89189E-15 | 1.28 | 0.60 | 0.44 | 0.76 | 54.30 | 42.28 | 12.02 |

[*] Stringent genome-wide significance threshold: p-value $<5\text{x}10^{-8}$.

achieved: AUC = 0.856 (0.749~0.963), sensitivity = 0.917 (0.917~1.000) and specificity = 0.708 (0.526~0.890) after 10-fold cross-validation. The logistic regression model is represented below:

$$\text{logit}(P) = \log(P/(1-P)) = -0.072 - 1.5 \; \text{cg04515524} - 1.901 \; \text{cg00613827} - 0.992$$
$$\text{cg02356786} - 1.358 \; \text{cg07509935},$$

where P is $\Pr(y = 1|x)$.

AI-based analysis, and in particular DL, was superior to conventional regression analysis, Tables 4 and 5, S3–S6 Tables. Overall, these results appear to support the robustness of blood-based epigenomic markers for AD prediction.

## Network and pathway analyses results

The network and pathway analysis based on intragenic epigenomic markers identified significantly enriched canonical pathways. The molecular pathways that were found to be statistically significantly overrepresented were Cardiac Hypertrophy Signaling, Sirtuin Signaling, FGF Signaling, Wnt/β-catenin Signaling, and Neuregulin Signaling (S7 Table). The over-represented

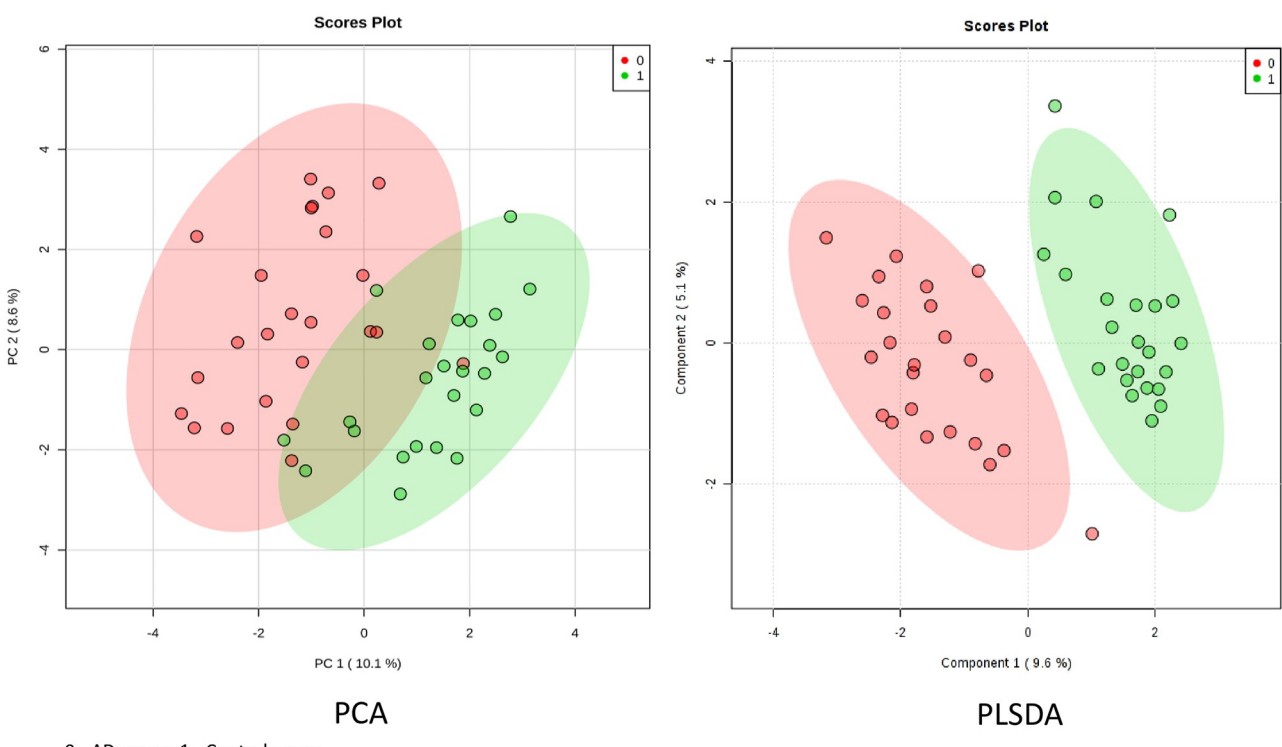

PCA                    PLSDA

0 - AD group; 1 - Control group

**Fig 1. Principal Component Analysis (PCA) and Partial Least Square Discriminant Analyses (PLS-DA) with intragenic markers.**

disease pathways were Abnormal morphology of the cerebral cortex, Gliosis, Hydrocephalus, Morphology of nervous system, Ventricular hypertrophy, dilated cardiomyopathy, and Inflammatory response (S8 Table). The related gene (Fig 2) and disease pathways (Fig 3) are depicted. S9 Table provides a summary of genes that were significantly differentially methylated and plausibly linked to AD development.

To evaluate the correlation between leukocyte methylation and gene expression in the brain, we matched our result with the study of Miller et al., [36] They reported the genes that

**Table 4. Alzheimer's disease prediction based on all intragenic* CpG markers only.**

|  | SVM | GLM | PAM | RF | LDA | DL |
|---|---|---|---|---|---|---|
| AUC 95% CI | 0.9898 (0.8000–1) | 0.9880 (0.8000–1) | 0.9877 (0.8000–1) | 0.9620 (0.8000–1) | 0.9325 (0.8000–1) | 0.9920 (0.8000–1) |
| Sensitivity | 0.9100 | 0.9500 | 0.9200 | 0.9100 | 0.9000 | 0.9750 |
| Specificity | 0.9700 | 0.9800 | 0.9400 | 0.9500 | 0.9000 | 0.9700 |

* based on analysis of 283,143 CpG loci.

Important predictors in order.

SVM: cg10304803, cg07589235, cg09991306, cg07773593, cg11035296.

GLM: cg02434121, cg27066201, cg14185918, cg07079724, cg04898026.

PAM: cg25179758, cg08086084, cg21027526, cg17840509, cg24644672.

RF: cg25179758, cg27066201, cg14185918, cg07773593, cg11035296.

LDA: cg09991306, cg07773593, cg27066201, cg14185918, cg24644672.

DL: cg10304803, cg07589235, cg09991306, cg07773593, cg11035296.

Support Vector Machine (SVM), Generalized Linear Model (GLM), Prediction Analysis for Microarrays (PAM), Random Forest (RF), Linear Discriminant Analysis (LDA), and Deep Learning (DL).

**Table 5. Alzheimer's disease prediction based on intergenic (extragenic) CpG markers* only.**

|  | SVM | GLM | PAM | RF | LDA | DL |
|---|---|---|---|---|---|---|
| AUC 95% CI | 0.9970 (0.8000–1) | 0.9980 (0.8000–1) | 0.9977 (0.8000–1) | 0.9820 (0.8000–1) | 0.9725 (0.8000–1) | 0.9990 (0.8000–1) |
| Sensitivity | 0.9200 | 0.9400 | 0.9300 | 0.9200 | 0.9200 | 0.9750 |
| Specificity | 0.9860 | 0.9810 | 0.9580 | 0.9550 | 0.9100 | 0.9750 |

*—analysis based on 244,246 markers.

Important predictors in order.

SVM: cg01941243, cg09301498, cg27128435, cg03043243, cg09050832.

GLM: rs4331560, cg15410835, cg05477405, cg16818568, cg01938825.

PAM: cg19008148, cg02875416, cg18232989, cg25761791, cg06842409.

RF: cg19008148, cg15410835, cg05477405, cg03043243, cg09050832.

LDA: cg15410835, cg27128435, cg03043243, cg25761791, cg06842409.

DL: cg01941243, cg09301498, cg27128435, cg03043243, cg09050832.

Support Vector Machine (SVM), Generalized Linear Model (GLM), Prediction Analysis for Microarrays (PAM), Random Forest (RF), Linear Discriminant Analysis (LDA), and Deep Learning (DL).

were differentially expressed in the CA1 and CA3 regions of the brain from AD patients. We found 13 genes differentially expressed in CA1 and CA3 regions of the brain from that study [36] were significantly differentially methylated in circulating leukocytes. These were *CCDC3*, *CPS1*, *ERMAP*, *FAM84B*, *MIB2*, *PTPRC*, *SARM1*, *SEC11A*, *TRIM6*, *TXNIP* found to be differentially expressed in the CA1 region and *ADM*, *ANKS1B*, *LANCL1* differentially expressed in the CA3 region [36]. Among these, *CPS1* is involved in ammoniac intake in the urea cycle [37], *PTPRC* is one of the microglial expressed gene [38], *SARM1* is involved in axon degeneration, which a factor observed in AD [39], *TXNIP* is linked to neuroprotective function [40], *ANKS1B* regulates hippocampal synaptic transmission [41] and *LANCL1* is required for normal neuronal function [42]. We also compared our methylation results with a previous study evaluating differentially methylated genes in leukocyte blood samples of mono and dizygotic

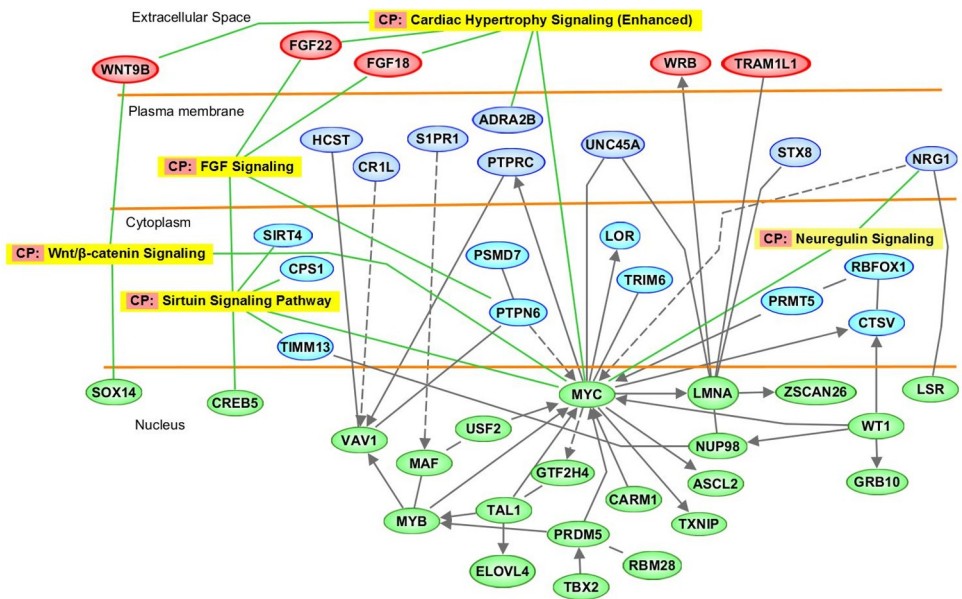

**Fig 2. Epigenetically dysregulated molecular pathways in AD.**

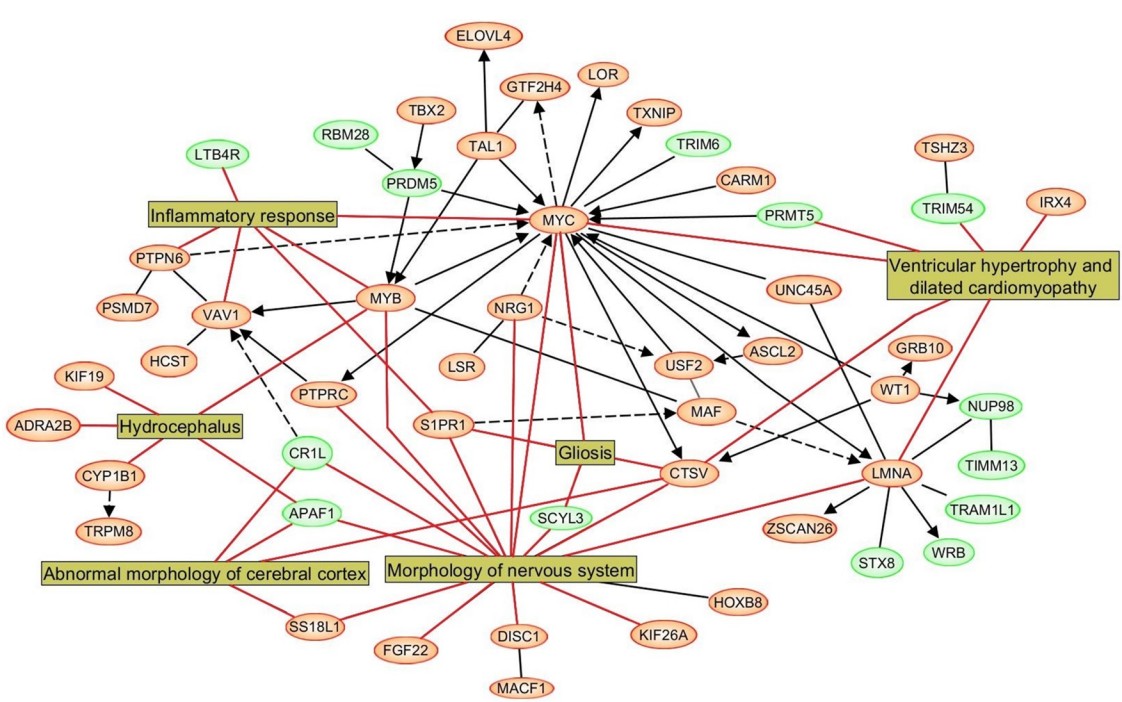

**Fig 3. Epigenetically dysregulated disease pathways in AD.**

twins [43]. These twin pairs were discordant for methylation. Twenty-two of those differentially methylated genes were also found to be significantly differentially methylated in our study. The direction i.e. increased versus decreased, of methylation change was similar in that and the current study for the following genes: *C5orf38*, *CDK20*, *CREB5*, *CTSV*, *DISC1*, *ELOVL4*, *FGF22*, *HOXC12*, *IGSF21*, *IGSF9B*, *IRX4*, *MAF*, *S1PR1*, *STX8*, *TBX2*, and *TSHZ3*. However, for genes *ASCL2*, *FAM124B*, *FAM174B*, *KIF19*, *KIF26A*, and *WSCD1* both studies found significant methylation changes in the leukocyte DNA of AD cases however the direction of the methylation change was discordant between the studies [43].

## Discussion

Dementia represents a looming global health crisis. The problem is expected to worsen with an anticipated explosion in the aged population in the future [44]. The direct health care costs, along with intangible costs, are burdensome at an estimated $550 billion annually [45]. The inpatient hospital cost for individuals 65 years and over with Alzheimer's and other dementias is greater than 3 times that of similarly aged individuals without dementia, with the nursing home facility costs greater than 20 times that of the latter group [46]. Despite the current absence of curative therapy, the justification for biomarker development remains compelling. Early detection of AD is needed to ensure early interventions that could potentially mitigate disease severity and also give families time to better prepare for the care of such individuals. With a very active drug pipeline, early detection will be needed to identify appropriate candidates for these trials. Finally, early detection and resulting intervention to slow disease progression could minimize time spent with severe dementia and promote the preservation of cognitive function for as long as possible. This would be beneficial for quality of life [47] and health care costs considerations. AD is a slowly developing disorder enhancing the feasibility of achieving these objectives.

Consistent with the call for the integration of breakthrough technologies (systems biology, genomics, big data science, and blood-based markers) to advance precision medicine objectives in AD [48], we combined AI analysis with leukocyte epigenomic data for AD prediction. Using raw intragenic CpG markers alone, we achieved a highly accurate prediction of AD using ML-based techniques. All the AI platforms achieved an AUC ≥0.93 using leukocyte epigenomic data. In the case of Deep Learning, we obtained an AUC = 0.99 with 97% sensitivity and specificity values. Additionally, we achieved high predictive accuracy using intergenic/extragenic CpG sites alone for AD detection. The use of conventional clinical predictors and MMSE did not improve performance further.

AI is superior to conventional statistical tools for the analysis of big data generated by omics analysis [17,49]. It is a powerful tool for discriminating and classifying groups. It can identify multiple markers each with limited individual predictive capabilities which when combined achieve excellent discriminating performance. To minimize the chances of overfitting strategies such as RF were used (see Supplementary Methods). For the sake of comparison, we also investigated the predictive performance of conventional logistic regression. Employing cross-validation techniques, regression analysis yielded good predictive accuracy for AD based on methylation markers: AUC (95%CI) = 0.85 (0.74–0.96) but less than that of AI. This, however, further supports the robustness of the leukocyte epigenomic markers for AD detection.

Currently, a range of imaging markers continues to be deployed in clinical and research diagnosis and evaluation of AD. These include CT, MRI, and PET imaging of the brain and CSF amyloid and tau levels. A systematic review of imaging biomarkers revealed that currently, the most commonly utilized antemortem diagnostic tests have achieved moderate to good diagnostic accuracy [50]. The expense, and in some cases the invasive nature of these tests, precludes use in the general aged population. Psychological testing including the MMSE, the most widely used cognitive test, might not be readily available in many primary care settings where the majority of elderly patients receive clinical care. Further, the MMSE was found on meta-analysis to have only modest accuracy for ruling out dementia when deployed in a community or primary care settings [51]. Based on all these considerations, there remains a need for accurate biological screening tests in a low to moderate risk setting.

While not a requirement, an important collateral benefit of an ideal biomarker, beyond predictive accuracy, is the ability to help elucidate disease pathogenesis. We identified altered CpG methylation in several individual genes (*CR1L*, *MYC*, *NRG1*, *LMNA*, *ELOVL4*, *MYB*, *AGPAT1*, and *NSG1*) previously reported playing a role in AD. Single nucleotide polymorphisms in these genes increase AD risk by affecting the formation of neurofibrillary tangles, neuronal apoptosis, and neuronal vesicle trafficking in AD (S7 Table). [52–60] Further, IPA found enrichment of several pathways involved in brain and neuronal development and brain and cardiovascular function such as abnormal morphology of cerebral cortex, gliosis, the morphology of the nervous system, Inflammatory response and cardiac ventricular hypertrophy, and dilated cardiomyopathy (Figs 2 and 3 and S5–S7 Tables).

AD appears to primarily affect the medial temporal cortex of the brain and both AD and aging affect the inferior parietal lobe and dorsolateral prefrontal cortex regions of the brain [61]. The accumulation of a significant volume of neurofibrillary tangles in the neocortical region is a hallmark of AD development [62]. We found significant epigenetic changes in genes (*CR1L*, *CTSV*, *APAF1*, and *SS18L1*) responsible for cerebral cortical morphology.

Microglia are immune cells residing in the brain. Proliferation and hypertrophy of these cells (gliosis) occur in response to CNS damage. Gliosis can lead to neuroinflammation and induce tau pathology thus accelerating neurodegeneration. In the case of AD, amyloid-β plaque deposition aggravates gliosis [63]. Our pathway analysis suggested a relationship between

abnormal methylation and increased gliosis in AD. *S1PR1* and *MYC* genes were hypermethylated in our study. The *S1PR1* gene is involved in CNS inflammation [64] and the *MYC* gene in astrogliosis and inflammatory response [65].

We also found an over-representation of molecular pathways, including cardiac hypertrophy signaling and Wnt signaling, in AD. Vascular disease is strongly associated with negative effects on cognition [66]. Left ventricular hypertrophy is reported to be an independent risk factor for dementia [67]. We identified genes involved in cardiac hypertrophy signaling that displayed altered methylation in the AD group. Polymorphisms of the *ADRA2B* gene have been linked to cerebrovascular disorders [68]. The *FGF18* and *FGF22* genes are known to play a role in heart development and physiological processes [69] while the *MYC* gene is implicated in angiogenesis, cardiomyogenesis, apoptosis, oxidative stress response and plays a major role in initiating and maintaining cardiac hypertrophy and contractility [70]. In our study, these genes were found to be significantly differentially methylated and further support an important link between cardiovascular function and AD.

The Wnt/β-catenin signaling pathway is one possible link between cardiovascular disease and dementia. Wnt signaling is critical for the developmental processes in multiple organs including that of the heart. The pathway is reactivated in many post-natal cardiac disorders [71]. The activation of Wnt signaling has a neuroprotective effect while inhibition promotes neurodegeneration [72]. Downregulated Wnt/β-catenin signaling is associated with AD [73]. Wnt/β-catenin signaling genes such as *MYC*, *SOX14*, and *WNT9B* were found to be hypermethylated in the study.

A limitation of our study was the relatively small sample size. We also performed bootstrapping to confirm the stability of our estimates (see Supplemental Methods section). This slightly increased the performance estimates for 4 platforms including DL while slightly decreased the performance in 2 AI platforms. We intend to perform follow-up validation studies in a larger cohort of patients. Despite the study size, we demonstrated highly significant methylation changes in circulating leukocytes in AD. Highly accurate AD prediction was observed using an AI platform and different marker combinations. Also, while expression studies were not performed in this particular analysis, several CpG site methylation differences in AD cases versus controls were greater than 5–10%. This level of methylation difference has been noted to correlate with changes in corresponding gene expression [74]. While we did not perform expression analysis in the current study, we did find evidence of significant methylation changes in some leukocyte genes that have been previously reported to be differentially expressed in AD brains [36]. These findings also help to validate our data.

While significant epigenetic changes were also identified in the intergenic/ extragenic sites, we are currently unable to report the specific mechanisms of their contribution to AD pathogenesis as these sites have not been linked to particular genes. It is known however that intergenic/extragenic sites can exert long-range influence and control gene function.

Overfitting can be a challenge with AI analysis. To avoid overfitting in the DL model strategies including the use of regularization parameters, dropout, and controlling the input- dropout ratio were used and are detailed in the Supplemental Methods section. For the other AI platforms, several parameters were used to tune the models and to overcome the overfitting problem: number of trees for RF, classification cost for SVM, and threshold amount for shrinking toward the centroid for PAM.

Another limitation of the study is that we were not able to eliminate the possibility that some of the observed epigenetic changes were not due to co-morbidities such as schizophrenia, bipolar disorder, or epilepsy. Given the age of the study subjects, co-morbidities are the norm rather than the exceptions in AD. We did not however identify significant differences in the frequency of these disorders in our AD versus control groups. We did not have access to the

medications of our study group. The study included a higher percentage of females in both the case and control groups. This however is consistent with the distinct gender-based demographics of the disorder. There was however no significant difference in the gender ratios of the case and control groups. Further, we removed all probes associated with X and Y chromosomes to minimize gender bias. We have excluded any CpGs having close association (0 to 10 bp distance) with single nucleotide polymorphisms to avoid genetic mutational association with the methylation changes. Finally, no information on the APOE gene mutation status was available for this particular cohort. These are not routinely obtained in the assessment of our clinical patients.

A significant strength of our study is the novelty, i.e. the use of blood leukocytes to accurately detect AD and also for interrogating the pathogenesis of AD. Leukocyte samples are easily obtained, raising the prospect of a minimally invasive and potentially affordable technique for investigation of the mechanisms, detection, as well as longitudinal monitoring of AD. The potential value of methylation changes in blood leukocytes for the detection of brain disorders including schizophrenia has been previously reported [75,76]. Of interest, we did find overlap in some of the genes that were significantly differentially methylated in AD in our study and a prior report of leukocyte DNA methylation variation in twins discordant for AD [43]. This provides further validation to the use of leukocyte methylation for the investigation of AD.

In summary, we have performed genome-wide methylation analysis in blood leucocytes and identified significant methylation changes in genes, gene networks, and disease pathways that were previously known or suspected to play an important role in AD. Significant methylation changes were also found in intergenic i.e. extragenic sites. Using AI techniques, highly accurate leukocyte epigenomic prediction of AD was reported for the first time to the authors' knowledge. The results could potentially advance the precision medicine objectives that have been outlined for AD [48]. Our work provides evidence in support of the view that epigenetic factors may play a pivotal role in AD development. Further validation studies using a larger number of subjects are necessary to confirm and expand on our findings.

## Supporting information

**S1 Table. Clinical and demographic characteristics: AD compared to unaffected control subjects.**
(DOCX)

**S2 Table. Remaining (119 among 152) differentially methylated significant intragenic CpG markers.**
(DOCX)

**S3 Table. Alzheimer's disease prediction based on intragenic CpG markers.**
(DOCX)

**S4 Table. Alzheimer's disease prediction based on intergenic/extragenic CpG markers.**
(DOCX)

**S5 Table. Alzheimer's disease prediction based on intragenic CpG markers only: Genome-wide significance threshold*.**
(DOCX)

**S6 Table. Alzheimer's *disease prediction based on* intergenic/extragenic CpG markers (stringent* significance threshold).**
(DOCX)

**S7 Table. Differentially methylated genes enriched under molecular pathways in Alzheimer's disease (Ingenuity pathway analysis).**
(DOCX)

**S8 Table. Differentially methylated genes enriched in disease pathways of Alzheimer's disease (Ingenuity pathway analysis).**
(DOCX)

**S9 Table. List of few genes that were found to be significantly differentially.**
(DOCX)

**S1 File.**
(DOCX)

## Author Contributions

**Conceptualization:** Ray O. Bahado-Singh, Raghu P. Metpally, David J. Carey, Halil Bisgin, Uppala Radhakrishna.

**Data curation:** Ray O. Bahado-Singh, Sangeetha Vishweswaraiah, Buket Aydas, Ali Yilmaz, Raghu P. Metpally, David J. Carey, Wade H. Berrettini, George D. Wilson, Stewart F. Graham, Uppala Radhakrishna.

**Formal analysis:** Sangeetha Vishweswaraiah, Buket Aydas, Ali Yilmaz, Raghu P. Metpally, David J. Carey, Richard C. Crist, George D. Wilson, Khalid Imam, Michael Maddens, Stewart F. Graham, Uppala Radhakrishna.

**Funding acquisition:** Ray O. Bahado-Singh, Raghu P. Metpally.

**Investigation:** Ray O. Bahado-Singh, Raghu P. Metpally, Richard C. Crist, Wade H. Berrettini, Khalid Imam, Stewart F. Graham, Uppala Radhakrishna.

**Methodology:** Sangeetha Vishweswaraiah, Buket Aydas, Ali Yilmaz, Raghu P. Metpally, Michael Maddens, Halil Bisgin, Uppala Radhakrishna.

**Project administration:** Ray O. Bahado-Singh, Raghu P. Metpally, Khalid Imam, Uppala Radhakrishna.

**Resources:** Ray O. Bahado-Singh, Ali Yilmaz, Uppala Radhakrishna.

**Software:** Sangeetha Vishweswaraiah, Buket Aydas, Ali Yilmaz, Raghu P. Metpally, David J. Carey, Richard C. Crist, Halil Bisgin, Uppala Radhakrishna.

**Supervision:** Ray O. Bahado-Singh, Buket Aydas, Raghu P. Metpally, Wade H. Berrettini, George D. Wilson, Michael Maddens, Halil Bisgin, Stewart F. Graham, Uppala Radhakrishna.

**Validation:** Sangeetha Vishweswaraiah, Buket Aydas, Ali Yilmaz, Raghu P. Metpally, David J. Carey, Richard C. Crist, Wade H. Berrettini, Halil Bisgin, Stewart F. Graham, Uppala Radhakrishna.

**Visualization:** Richard C. Crist, George D. Wilson, Khalid Imam, Michael Maddens, Halil Bisgin, Uppala Radhakrishna.

**Writing – original draft:** Ray O. Bahado-Singh, Sangeetha Vishweswaraiah, Buket Aydas, Ali Yilmaz, Raghu P. Metpally, David J. Carey, Richard C. Crist, Wade H. Berrettini, George D. Wilson, Khalid Imam, Michael Maddens, Uppala Radhakrishna.

**Writing – review & editing:** Ray O. Bahado-Singh, Sangeetha Vishweswaraiah, Buket Aydas, Raghu P. Metpally, David J. Carey, Richard C. Crist, George D. Wilson, Khalid Imam, Michael Maddens, Halil Bisgin, Stewart F. Graham, Uppala Radhakrishna.

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
