## [Decision Letter · Decision Letter 0]

25 Feb 2021

Artificial Intelligence and Leukocyte Epigenomics: Evaluation and Prediction of late-onset Alzheimer's disease

PONE-D-21-04089

Dear Dr. Uppala,

We’re pleased to inform you that your manuscript has been judged scientifically suitable for publication and will be formally accepted for publication once it meets all outstanding technical requirements.

Kind regards,

Udai Pandey, PhD

Academic Editor

PLOS ONE

Journal Requirements:

1. Thank you for stating the following in the Financial Disclosure section: 'No'

We note that one or more of the authors are employed by a commercial company: Meridian Health Plans

c. Please respond by return email with an updated Funding Statement and Competing Interests Statement and we will change the online submission form on your behalf.

Reviewers' comments:

Reviewer's Responses to Questions

**Comments to the Author**

1. Is the manuscript technically sound, and do the data support the conclusions?

Reviewer #1: Yes

Reviewer #2: Yes

2. Has the statistical analysis been performed appropriately and rigorously? 

Reviewer #1: Yes

Reviewer #2: Yes

3. Have the authors made all data underlying the findings in their manuscript fully available?

Reviewer #1: Yes

Reviewer #2: Yes

4. Is the manuscript presented in an intelligible fashion and written in standard English?

Reviewer #1: Yes

Reviewer #2: Yes

5. Review Comments to the Author

Reviewer #1: Bahado-Singh et al., have explored the increasing interest in the potential of epigenetic biomarkers to differentiate individuals with late onset Alzheimer's disease (LAOD) to those without. Some of the gene regions, and identified pathways appear biologically relevant, suggesting that these may not only serve as biomarkers but potentially involved in the disease pathogenesis. The main strength of this study is the robust systematic data driven analysis framework with extensive comparison of the methods, performance measures, and validation.

There still remains a gap between methylation biomarker studies and the potential to be used in a clinical setting. To date, almost all such studies have been case-control approach, and indicates the lack of temporal information to track disease incidence, and follow large groups of over time. As such it remains to be elucidated whether these methylation markers mediate detectable transcriptomic changes which will fully emphasize their efficacy.

This study indicates the potential for a future study, and validation of the findings in a well-defined longitudinal cohort consisting of LOAD cases, presymptomatic, and healthy controls.

Reviewer #2: This is a well written manuscript with important conclusions. The sample size is not extensive but still the analysis took into consideration this caveat. The use of AI provides high resolution analysis of the data.

6. PLOS authors have the option to publish the peer review history of their article (what does this mean?). If published, this will include your full peer review and any attached files.

Reviewer #1: No

Reviewer #2: No

---

## [Editor Report · Acceptance letter]

4 Mar 2021

PONE-D-21-04089 

Artificial Intelligence and Leukocyte Epigenomics: Evaluation and Prediction of late-onset Alzheimer's disease 

Dear Dr. Radhakrishna:

I'm pleased to inform you that your manuscript has been deemed suitable for publication in PLOS ONE. Congratulations! Your manuscript is now with our production department. 

Kind regards, 

on behalf of

Dr. Udai Pandey 

Academic Editor

PLOS ONE